# Translating Treg Therapy for Inflammatory Bowel Disease in Humanized Mice

**DOI:** 10.3390/cells10081847

**Published:** 2021-07-21

**Authors:** Sushmita Negi, Sheetal Saini, Nikunj Tandel, Kiran Sahu, Ravi P.N. Mishra, Rajeev K. Tyagi

**Affiliations:** 1Biomedical Parasitology and Nano-Immunology Lab, Division of Cell Biology and Immunology, CSIR-Institute of Microbial Technology (IMTECH), Chandigarh 160036, India; negisush3@imtech.res.in (S.N.); ssaini@imtech.res.in (S.S.); kirankumariroyal@gmail.com (K.S.); 2BERPDC Department, CSIR-Institute of Microbial Technology (IMTECH), Chandigarh 160036, India; 3Institute of Science, Nirma University, Ahmedabad, Gujarat 382481, India; nikunj.tandel@nirmauni.ac.in

**Keywords:** humanized mice, inflammatory bowels disease, Crohn’s disease, ulcerative colitis, human immune system, regulatory T cells

## Abstract

Crohn’s disease and ulcerative colitis, two major forms of inflammatory bowel disease (IBD) in humans, afflicted in genetically predisposed individuals due to dysregulated immune response directed against constituents of gut flora. The defective immune responses mounted against the regulatory mechanisms amplify and maintain the IBD-induced mucosal inflammation. Therefore, restoring the balance between inflammatory and anti-inflammatory immunepathways in the gut may contribute to halting the IBD-associated tissue-damaging immune response. Phenotypic and functional characterization of various immune-suppressive T cells (regulatory T cells; Tregs) over the last decade has been used to optimize the procedures for in vitro expansion of these cells for developing therapeutic interventional strategies. In this paper, we review the mechanisms of action and functional importance of Tregs during the pathogenesis of IBD and modulating the disease induced inflammation as well as role of mouse models including humanized mice repopulated with the human immune system (HIS) to study the IBD. “Humanized” mouse models provide new tools to analyze human Treg ontogeny, immunobiology, and therapy and the role of Tregs in developing interventional strategies against IBD. Overall, humanized mouse models replicate the human conditions and prove a viable tool to study molecular functions of human Tregs to harness their therapeutic potential.

## 1. Introduction

Inflammatory bowel disease (IBD) is a complex inflammatory chronic and pathological condition that includes Crohn’s disease (CD) and Ulcerative colitis (UC). CD primarily affects the small and large intestine whereas the prime targets for UC are colon and rectum [1]. Clinical manifestations of IBD are characterized by abdominal pain, rectal bleeding, bloody stools, tenesmus, diarrhea, weight loss, and the urgency to evacuate [1,2]. Pathogenesis of IBD involves various environmental, genetic and bacterial factors with dysregulated mucosal immune-mechanisms resulting in the disrupted intestinal homeostasis [3,4], and dysregulated mucosal immune response provokes robust inflammatory response against intestinal flora [2,5].

Immune sentinel subsets of CD4+ T cells such as Th (T helper cells)-1, Th2, Th17, and regulatory T cells (Tregs) play a crucial role in the pathogenesis of IBD. Immunological balance between effector Th cells and Trges is essential for maintaining immune-homeostasis [6,7]. Immunoregulatory Trges are characterized by the expression of transcription factor Forkhead box P3 (Foxp3), and surface marker CD25 [8,9], and are functionally immunosuppressive & important for immune tolerance [10]. Therapeutic arrangement based on Tregs is important to address the systemic inflammatory and autoimmune diseases such as IBD and rheumatoid arthritis [11,12,13]. Further, decreased number of Trges was seen in the patients with IBD than healthy control [14,15,16]. Inhibited generation of functionally impaired Trges contribute to the intestinal inflammation leading to colitis and other biological complications [17,18]. Functionally immune-suppressive Trges have reportedly shown to ameliorate the IBD-induced immune responses [19,20]. Reduced intestinal inflammation drives the control of IBD pathogenesis [21], and increased number of functional Tregs experimentally confirmed their “therapeutic potential” during IBD pathogenesis [22,23,24].

Present article is an orchestrated attempt to validate the functional aspects of Trges during the IBD pathogenesis for developing interventional therapeutic strategies. Further, role of Trges in the maintenance of peripheral immune tolerance with an emphasis on mucosal immunity has been discussed at length. Additionally, various transgenic and immunodeficient mice repopulated with the human immune system; mouse-human chimeric models and their importance to study of IBD pathogenesis and associated biological phenomena have been discussed.

## 2. IBD Pathogenesis

The epithelial layer of the human gut consists of goblet cells, columnar cells, paneth cells, endocrine cells, M cells, tuft cells, and epithelial resident intestinal stem cells. These cells are responsible for the differentiation of gut microbiota and secretion of various mucus-containing antimicrobial peptides [25,26,27]. The intestinal barrier contains innate immune cells such as dendritic cells (DCs), neutrophils, macrophages and innate lymphoid cells (ILCs) which reside in the state of hypo-responsiveness in a healthy human gut [28,29]. The mucosal macrophages prevent the inter-conversion of Th1 and Th17 cells by producing anti-inflammatory cytokines and thus promoting the differentiation of Trges [30,31,32]. The immune cells’ balance in intestinal mucosa and luminal content is crucial for the normal functioning of the mucosal immune system since dysregulated immune effecters result in the IBD pathogenesis [33,34,35]. The impaired innate immune system is responsible for the functional abnormalities of the adaptive immune system, and interconversion of effector Th cells and Trges causing IBD pathogenesis [35,36]. Therefore, balanced gut mucosal immunity to maintain immune homeostasis and protection is inevitably critical to fighting IBD.

Activated lamina propria (intestinal mucous membrane) secrete a large number of soluble immune mediators, including pro-inflammatory cytokines viz. tumor necrosis factor (TNF), interferon-gamma (IFN-γ), Interleukin (IL)-6, IL-12, IL-21, IL-23, IL-17 and anti-inflammatory cytokines such asIL-10, transforming growth factor (TGF-β) and IL-35in local tissues [37]. The imbalance between these secreted soluble mediators especially inflammatory and anti-inflammatory cytokines secreted by immune system results in IBD pathogenesis [3,37,38,39] (Figure 1). The exogenous administration of TNF and TNF-like cytokine 1Acytokines (TLA1) regulates the balance between Th1 and Th17 cell population in the inflamed colonic tissues [40]. Further, TLA1 modulates Foxp3 expression in Tregs and its function, and murine model of colitis has seen the alleviation of colitis when treated with Tregs expressing low levels of TLA1.TLA1 may promote the maintenance of Treg suppressor function in a death domain receptor 3 (DR3) dependent manner [41]. Passive administration of anti-TLA1 antibodies prevents the development of 2,4,6-trinitrobenzene sulfonic acid (TNBS)-induced colitis in mice, partially improves dextran Sulfate sodium (DSS)-induced colitis and decreases the intestinal fibrosis in a chronic colitis model [42,43,44].

A meta-analysis has identified more than 200 genomic loci associated with IBD pathogenesis, and 68% of loci have been shared by the UC and CD disease [45]. Genome-wide association studies (GWAS) have helped the scientific community to find out genes responsible for IBD pathology and nucleotide-binding oligomerization domain 2 (*NOD2*) is the first gene found to be associated with IBD [46]. The associated mutations in *NOD2* and polymorphisms within autophagy-related 16-like 1 (*ATG16L1*) genes have disseminated the pivotal role of autophagy in the pathogenicity of IBD [47,48,49]. NOD2 knockouts exhibited excessive intestinal inflammation as compared to NOD2 sufficient mice [50], and selective deletion of ATG16L1 in T cells ends with spontaneous intestinal inflammation, characterized by a decrease in Foxp3^+^ Tregs and aberrant expression of Th2 cells [51].

The contribution of gut homing or migration associated molecules such as α4b7 integrin, αEb7 integrin, CD62L, chemokine receptor (CCR)-4 (CCR4), CCR5, CCR7 and CCR9 in the pathogenesis of IBD is well known [52]. Defective or loss of expression of these molecules leads to the impaired Tregs trafficking to the target organs and thus inducing IBD. Loss of CCR7 and CCR4impairs the functions of Tregs in experimental colitis [53,54], and further CCR7 regulates the balance between Th1, Th17, and Trges in Crohn’s-like Murine ileitis [55]. b7 integrin deficiency impaired Tregs homing in IL-10 deficient mice and spontaneously increased IBD-induced inflammation [56]. In essence, all findings indicate the crucial role played by the number of functional Tregs since a compromised number of Tregs contribute to the pathogenesis of IBD and other associated biological impairments.

## 3. Regulatory T Cells (Tregs)

Tregs are heterogeneous cell populations of CD4+ T cells and possess immunosuppressive attributes. CD4+ Trges expressing high levels of IL-2 receptor α chain (CD25) and master transcription factor Foxp3 are the best-characterized populations with the immunosuppressive phenotype [57]. Foxp3 is essentially required for maintaining their immunosuppressive activity against infections, tumors, intestinal inflammation, allergy, and autoimmunity [8,9,10,58]. The absence of CD127 (IL-7 receptor α-chain) is considered as another feature of Tregs and up-regulates CD25 and Foxp3 expression upon activation. Thus a lower expression of CD127 is important alongwith elevated CD4, CD25, and Foxp3 expression to confirm the functional phenotype of immunoregulatory T cells [59,60]. Two main subsets of Trges are characterized as Foxp3 positive Trges and Foxp3 negative type 1 Treg (Tr1) cells.

### 3.1. tTregs and pTre

Depending on generation, Foxp3^+^ Trges are further categorized as naturally occurring thymus-derived Treg (tTreg) cells and Trges developed from conventional CD4+ T cells in the periphery (pTreg). These cells possess immunosuppressive functions and maintain peripheral tolerance [61,62]. tTregs are produced by the thymus at an early stage after birth and maintain tolerance toward self-antigens [63,64]. TGF-β1 directly enhances the Foxp3 promoter and encourages the generation of tTregs [65]. Besides, exposure of naive T-cells to its cognate-antigen leads to the differentiation of pTregs under tolerogenic conditions [66,67,68], and differentiation of pTregs is facilitated by the higher concentrations of TGF-β and higher levels of Foxp3 [69,70,71]. Therefore, TGF-β1 is a key cytokine and plays a crucial role in the differentiation of both subsets of Trges. The induction of Foxp3 in peripheral naive T cells is achieved by a higher concentration of TGF-β, retinoic acid, and CD28 co-stimulation [70,72,73,74]. Both, tTreg and pTregs show similar expression levels of FoxP3, CD25, CTLA-4, GITR, ICOS, CD103, CD127 and a broad T-cell receptor (TCR) repertoire to deploy various suppressive mechanisms to control effector cells [8,59,75,76]. Foxp3^+^ Trges are also known to secrete IL-10, TGF-β, and IL-35 [77,78] along with granzyme A and B [79,80,81]. Furthermore, tTregs express higher levels of neuropilin-1(Nrp1), TF Ikzf2 (Helios), PD-1, and ecto nucleotidase CD73 than pTregs [82,83]. Helios and Nrp1 are considered as markers for tTregs since their greater expression is seen in tTregs as compared to pTregs [83,84,85]. Interestingly, under *in-vivo* conditions, pTregs could express helios [86], and a fraction of the human tTreg population did not express helios [87]. Moreover, tTregs are not differentiated based on helios and Nrp1 expression in mice [88]. Treg specific demethylated region (TSDR) is highly demthylated in tTregs, and partially demethylated along with an unstable expression of Foxp3 and CD25 in pTregs [69,89,90]. Apart from TSDR, Ig superfamily surface protein GPA33along with other Treg cell markers was recently used to identify Trges of thymus origin since this molecule is stably expressed on tTregs [91].

### 3.2. Type 1 T regulatory (Tr1) Cells

Tr1 cells are unique Foxp3^−^ regulatory T cells that develop in the periphery and secrete elevated levels of immunosuppressive cytokines such as IL-10 and TGF-β [92,93]. tTreg and pTregs constitutively express Foxp3 and CD25 but these markers are expressed by Tr1 cells only in the activated state [94]. Tr1 cells are characterized by the co-expression of surface markers, CD49b and LAG-3 [95], and can be distinct due to the cytokine expression of IL-2, IL-10, IFN-γ, IL-5 and Il-17 [94] as well as granzyme B and perforin via cell death mechanism [93]. And, experimental evidences have confirmed the immunosuppressive function of Tr1 cells mediated by IL-10 [96,97].

## 4. Role of Tregs in IBD

Trges play a vital role in maintaining gut immune homeostasis and regulate pro-inflammatory responses elicited by the adaptive and innate immune effectors [52]. Scurfy mouse strain showing the severe autoimmune phenotype with a genetic defect in the Foxp3 gene and inhibit the Tregs development and leads to the dysregulated activation of the gut immune system [98], which mounts inflammation primarily in the gut. Further, the mutation in the human Foxp3 gene leads to a rare autoimmune dysregulation, polyendocrinopathy, enteropathy, X-linked (IPEX) syndrome along with other severe autoimmune diseases including arthritis, diabetes, allergy and IBD. This syndrome was seen due to impaired immune response mediated inflammation [17,99,100]. Furthermore Foxp3 expressing Trges are essential for maintaining the balance at the intestinal mucosal surface because intestinal inflammation gets chronic with the decreasing number of Foxp3^+^ Tregs [101]. DSS induced colitis in mice showings pontaneous depletion of Foxp3^+^ Tregs leading to an increase in the disease severity [15]. However, adoptive transfer of Tregs in Treg depleted mice (DEREG) showed a decrease in the severity and improved tissue conditions of experimental colitis [15].

Although patients with IBD showed an increased number of Tregs in the inflamed intestinal mucosa than un-inflamed mucosal part [102,103,104]. The phenotype and function of Tregs present in the inflamed mucosa or periphery of IBD patients or in experimental animals differ from those present in peripheral lymphoid organs of healthy control. Patients with IBD showed an increase in the number of peripheral Th17 cells, and a reduction in the peripheral Trges [105]. While in some cases patients with IBD showed higher expression of Foxp3 along with elevated levels of pro-inflammatory cytokines including IL-17A, IL-1β and IL-6 [105]. Moreover, the highest frequency of Foxp3^+^ IL-17 T-cells (Th17 and Treg intermediate cells) was seen in the inflamed mucosal tissues of patients with IBD [106,107] (Figure 2).

Foxp3^+^ Trges express effector T cell-specific transcription factor retinoic acid receptor-related orphan receptor gamma t (RORγt) and differentiate into Th-17 cells. These cells inhibit the immunosuppressive function of Tregs in patients with IBD [108]. Further, Tregs upregulate the expression of T-bet and express pro-inflammatory cytokine IFN-γ. Regulatory T cells are characterized as IFN-γ-expressing Th1 like Tregs and mount intestinal inflammation in patients with IBD. Th1 like Trges evoke inflammation since the IFN-γ-expressing cells accumulate at the site of inflammation in CD and UC and contribute to the IBD pathogenesis [109]. The accumulation of IFN-γ expressing Th1 like Trges is also observed in the inflamed colon in the DSS-induced colitis animal model [109]. The acquisition of pro-inflammatory behavior of Tregs in IBD most likely contributes to the uncontrolled inflammation in vivo. Furthermore, Tregs suppress the colonic inflammation by downregulating Th1 and Th17 responses in the adoptive transfer model of colitis [110,111,112], and passively transferred Trges have shown the ability to control inflammatory lesions in the experimental model of IBD [113].

Trges have been seen to be involved in the tissue repair mechanism of the intestine. Among the different populations of intestinal Treg i.e., Tr1 cells have been shown to mediate repair of the intestinal mucosa that co-expressed Th2 and master transcription factor GATA binding protein 3 (GATA3). Therefore, elevated expression of IL-33R, ST2, and amphiregulin (AREG), an epidermal growth factor receptor ligand was reported [114]. Furthermore, these factors are generally involved in tissue repair and are phenotypically characterized and expressed by GATA3^+^ Tregs [114]. Tissue repair and protection of gut is rendered by the human Tr1 Trges suppress the proliferation of T effector cells, elicit TNF and IL-1β based innate immune response and secret IL-22 to regulate the repairing of epithelium and promote barrier function [115].

## 5. Therapeutic Role of Tregs in IBD

Several patients with IBD showed tolerance to the current therapeutic arrangements. Therefore, the need for developing effective, safe and novel therapies for IBD is both attractive and urgent. Newer and effective immunotherapies involving anti-TNF agents (infliximab, adalimumab, and certolizumab) have shown remarkable progress toreducethe need for surgery and hospitalization of IBD patients [116,117]. However, meta-analysis conducted by Ford et al., suggested that usage of anti-TNF agents may increase the risk of getting opportunistic infections in IBD patients [118]. Other recent reports also suggest the higher chance of getting infections such as histoplasmosis, aspergillosis and cytomegalovirus infection [119] and anti-TNF monotherapy was found to be responsible for higher mycobacterial and bacterial infection; however, in a combination therapy with thiopurine will increase the risk of getting serious infections [119,120]. The REFURBISH study was reported that the risk of getting T-cell non-Hodgkin’s lymphoma in IBD patients is higher during combinational therapy with compare to anti-TNF monotherapy [121] whereas the another cohort study delineate that even the ant--TNF monotherapy is associated with lymphoma formation in small number but have the higher statistical significant. And, it put on more on risk during the combinational therapy [122]. Other than this other paradoxical side effects such as psoriasis/psoriasiform skin, development of sarcoidosis-like lesions, late occurrence of arthritis/synovitis and lupus-like syndrome (0.5 to 1% of patients) can also be developed [123,124,125,126,127]. Additionally, novel therapies including JAK inhibitor [128,129,130], anti-MAdCAM-1 [131,132,133,134,135], an anti-SMAD7 antisense oligonucleotide (mongersen) [136,137,138], S1P1 [128,139] and anti-interleukin (IL)-12/23 (ustekinumab) [140,141,142,143,144,145,146] have been under investigation for safety and other purposes.

Recently, cellular therapies have been used as potential therapeutic strategies for IBD patients [147]. The role of Tregs in the preclinical models of colitis has been well understood, and recent investigations and phase 1 clinical trials have proven the safety and efficacy of Trges. A marked difference in the number of Tregs in patients (the inflamed mucosa or peripheral blood) and experimental animal models of IBD have been observed [101,148]. Experimental model of IBD showed an increase in Tregs percentage in the inflamed ileum with a reduced immunosuppressive function and IL-10 production. The dysregulated expression between Th17 cells and Tregs was also observed in UC animal model with downregulated mRNA expression of Foxp3 and IL-10 levels in Trges [149]. In UC patients, a significant increase of IL-17 and Th17 cells and a simultaneous decrease of TGF-β and Trges in serum as compared to healthy control was seen [150]. Th-17 cells and associated cytokines IL-17 and IL-23 were found to be decreased in patients with IBD along with a decrease in the Tregs and associated cytokines IL-10 and TGF-β [151]. IL-10 is known to induce Treg mediated suppression of Th-17 cells in a STAT-3 dependent manner [152]. The improvement in the clinical and histological parameters was observed when Trges were adoptively transferred in Rag^−/−^ or severe combined immunodeficient (SCID) mice [153]. Rapamycin-expanded Trges (Th cells cultured in presence of rapamycin) were shown to suppress colitis in SCID mice [154]. And, ovalbumin (OVA) induced Tregs from DO11.10 mice prevented colitis together with increased TGF-β and IL-10 secretion in SCID-bg mice [155]. The safety and efficacy of OVA-Treg therapy were assessed for refractory CD in an open-labeled multicenter phase I/II clinical trial. This study showed the dose-related efficacy because infusion of ova-specific Tregs treatment was well-tolerated, and 40%patients showed a reduced CD activity on 5 and 8-week post-treatment [156]. In vitro expanded CD45RA^+^ Tregs cells were shown to express stable Foxp3 locus, which enhanced their suppressive ability and prevented their conversion to Th17 phenotype in the SCID xenotransplant model [24]. Additionally, CD45RA^−^ and CD45RA^+^ expanded Tregs expressed a high level of gut homing receptor α4β7 integrin, CD62L, and CCR7 to facilitate their intestinal homing [24]. In an active CD mucosa, CD45RA^+^ Tregs healed the inflammation of lamina propria and mesenteric lymph nodes [24]. Tregs isolated from the lamina propria of active IBD patients and in experimental model (DSS induced colitis) express T-bet and IFN-γ (Th-1 like Tregs) and stimulates the early stages of inflammation. Further, T-bet KO showed the development of less severe colitis with the dysregulated Th1 immune response. It suggests that T-bet expression in Tregs is required for the development of colitis [109]. In the end, Treg immunotherapy with in vitro expanded Tregs (NCT03185000) for treating the Crohn’s disease (TRIBUTE trial) is underway.

## 6. Animal Models to Study IBD

Numerous animal models have been used to study many immune-effecters such as inflammatory mediators, chemokines and cytokines, pathogenic bacteria and effector Tregs. The experimental animal models of IBD produce the wealth of information to develop further understanding of the pathogenesis of IBD [52,157,158,159]. Genetically modified (GM) mice play an instrumental role to study of gastrointestinal (GI) tract disorders. GM mice including knockout, transgenic and mutant animals have proven a tool to unravel mechanisms underlying the intestinal inflammation and pathogenesis of systemic inflammatory diseases such as colitis and RA. Therefore, experimental induction of colitis in the mouse models has been developed by different chemicals including DSS, TNBS, oxazolone or acetic acid [100,160,161]. Furthermore, adoptive transfer models are being currently used and T-cell deficient mice reconstituted with Trges and depleted naïve T-cells were selected from congenic donor mice (Table 1) [52]. Moreover, few spontaneous and humanized mouse models have been developed to understand the pathogenesis to devising therapeutic treatments against colitis [162].

### 6.1. Bacteria-Infected Models

Studies carried out in the experimental mouse models of IBD suggested sensitivity of Treg cell compartment towards the changes occurred in the microbial environment [19]. Germ-free, pathogen-free, and gnotobiotic mice generally express a lower level of Foxp3 with reduced colonic Tregs compared to the Tregs extracted from wild-type mice. Treg population showed an increase with the decreased bacterial load upon receiving vancomycin treatment [164], and mice infected with *Citrobactor rodentium* were used to mimic the acute intestinal inflammation. Further, *Helicobacter*-infected mice are preferred as bacterial infection models over combination models to see the synergistic effect of drugs. Inflammatory signals emanating from bacterial DNA play an important role and inhibit the inducible Tregs, and their differentiation in Toll-like receptor 9 (TLR9) deficient mice showing a higher number of Tregs in small intestine [19,165]. Mice receiving DSS treatment showed a significant increase in the members of *Bacteroidaceae* and *Clostridium* spp. from *Bacteroides distasonis* and *Clostridium ramosum* families in the intestine [166]. Subsequent studies showed the elevated 16s rRNA gene copy numbers of mucin-degrading Gram-negative bacterium *Akkermansia muciniphilia* and *Enterobacteriaceae* to establish a correlation with the disease severity in DSS treated mice [167]. The increased number of microorganisms such as *Enterobacteriaceae* and adherent-invasive *E. coli* develops colitis in IL-10 deficient mice and provokes induction of inflammation leading to the cancer development [168,169]. Similarly, an increased number of *Bacteroides* and *Porphyromonas* genera in ApcΔ^468^/IL-10^−/−^ mice mounts inflammation and colon polyposis [170]. Furthermore, increased inflammation with the increased number of *Enterobacteriaceae* and *Bacteriodes* was seen in the TNBS induced colitis animals [171].

### 6.2. Genetically Modified Animal Models

A variety of gene knockout models are available to study the innate and adaptive immune responses elicited by the IBD pathogenesis or other intestinal infections. The modified genetic lines produce phenotypes that investigate the immunological aspects of intestinal infection and inflammation [52]. Commonly knockout (KO) genes used in the murine model of intestinal inflammation are IL-10, IL-23R, CD4+CD25+, NOD2/CARD15, TGF-β1, RAG, ATG16L1, APCmin/+, IL-2, TNF-α, STAT3, NFκB, Muc2, IFN-γ, MyD88 and TLR [163]. A reduced number of Tregs with compromised functional activity was observed due to the genetic modification by knocking out IL-2-/, IL-2R-/, and IL-10-/ genes in some immunodeficient mice to develop IBD models [52,172,173]. Innate or mucosal immunity-related gene-deficient mice such as NOD2-/, myeloid differentiation primary response 88 (MYD88)-/, nuclear factor-kB (NF-kB)^−/−^, cytokine-deficiency induced colitis susceptibility 1 (CDCS1)-/, multidrug resistance gene 1a (MDR1a)-/, and TLR5-/ showed inflammatory lesions in colon [52,174,175,176,177]. T-cell receptor TCR^−/−^ mice have been used to study various immunobiological phenomena taking place during IBD pathogenesis, and the development of IBD-like lesions are seen due to the over-expression of TNF-α and signal transducer and activator of transcription4 (STAT4) gene in TCR^−/−^ mice [178,179]. Overall, there are various gene-manipulated models wherein epithelial barriers and immune regulation-associated genes are manipulated by knockdown, knock-in, conditional knockout, or transgenic mice [52], and reduced Treg cell number and their impaired functions were observed in these animals. The different animal models used to study IBD pathogenesis is depicted in Figure 3.

### 6.3. Humanized Mouse Models

Characterization and pre-clinical therapeutic applications are being explored by studying the immune response of patients with IBD in the experimental animal model(s) without putting human life at risk. Evaluation of hematopoietic stem cells (HSCs) with mutations in Foxp3 and bone-marrow-derived CD34+ HSCs received from a patient with mutations and impaired function of IL-10Rshowed severe medical-refractory infantile-onset of IBD.HSCs administration inNOD.Cg-Prkdc^scid^Il2rg^tm1Wjl^/SzJ(NSG) mice lacking murine MHC-II and expressing HLA-DR1hardly showed the mounted intestinal inflammation as was seen in mice and humans carrying mutations in IL-10 or IL-10R genes [180,181]. The increased frequency of CD19+ B cells was assessed in spleen and mesenteric lymph nodes compared to the control in peripheral lymphoid cells reconstituted mice. Some patients with deleterious IL-10R mutations showed the development of B cell lymphoma and presented a barrier in assessing the potential role of this pathway in the regulation of B cell development [182,183]. Human immune cells recovered from reconstituted NSG mice were found non-responsive to the exogenous IL-10 treatment, and these observations were consistent with the results obtained by using peripheral blood mononuclear cells (PBMCs) from IBD patients [184]. Interestingly, NSG mice harboring transgene encoding human KITLG, GM-CSF, and IL3 (NSG-SGM3) injected with IL-10R1deficient PBMCs were seen susceptible to DSS-induced colitis compared to those receiving healthy control PBMCs. These experimental observations paved the ways to facilitate the development of therapeutic interventional approaches against patients with IL-10R mutations. Fully reconstituted immunodeficient mice with CD34+ HSCs isolated from patients with IL-10R mutations are not suitable for assessing the therapeutic biologics aiming at developing interventional approaches against IBD. Human immune system repopulated mice (humanized mice) would be appropriate to screen gene therapy-based approaches for restoring IL-10R signaling [184,185].

Interleukin-2 is a key cytokine that controls the differentiation, survival and function of Tregs [185,186,187]. Moreover, low dose IL-2 is known to activate Tregs in the peripheral blood and colonic lamina propria isolated from IBD patients in culture as well as HIS mice. And, Tyagi et al., 2021 explored the role of low dose IL-2 in expanding functional Tregs in HSC reconstituted NSG humanized mice [185]. NSG mice reconstituted with healthy donor PBMCs receiving rectal anema with TNBS on day 5 following immune reconstitution to induce colitis. Mice receiving low-dose (10K) IL-2 were shown to reduce the weight loss and histology scores compared to those receiving treatment with a higher (50K) dose of IL-2 [180,185].

Further, the percentage of Foxp3^+^ IL-10^+^ TGF-β^+^ natural Tregs, Foxp3^−^ IL-10^+^ TGF-β^−^ induced Tregs, CD127^−^ induced Tregs and CD8+ Tregs was measured at different time points in DSS-induced experimental colitis model in murine lamina propria lymphocytes, mesenteric lymph nodes and peripheral blood [148]. %age of Foxp3^+^ IL-10^+^ TGF-β^+^ natural Tregs show a decrease during chronic inflammation induced by IBD in humans and mice and proliferated significantly during remission. The intestinal inflammation exhibited a decrease in the percentage of CD8+ Tregs and remained lower in the remittent stage of human IBD. Only enhanced proliferation of lamina propria lymphocytes derived CD8+ Treg was reported on day 7in DSS-induced murine colitis. Furthermore, results suggest that Foxp3^+^ IL-10^+^ TGF-β^+^ natural Tregs might be crucial for the suppression and protection from immune-related mucosal injury during the chronic stages in IBD [148].

Local delivery of low numbers of human Treg by intradermal injection was shown to prevent skin inflammation in a humanized mouse model (huPBL-SCID-huSkin allograft model) [188]. A dose of only 1 × 10^5^ freshly isolated, non-expanded Tregs injected intradermally close to the transplanted human skin prevented the inflammation induced by the grafted tissue, and intraperitoneal injection of human allogeneic PBMCs and Tregs used were used as 400:1. Inhibition of epidermal thickening sustained Keratin-10 expression, absence of Keratin-16 up-regulation and prevention of human CD3+ T cell influx was observed following the cell administration [188]. Also, concomitant reduction of human T cells was observed in the lymph nodes and spleen of mice. Moreover, injection of Tregs at the contralateral side inhibits the skin inflammation which advocates for the reduction of local and systemic inflammation. In brief, local application of Treg might be an attractive strategy to suppress inflammation in vivo without requiring prior *ex-vivo* expansion [188]

The simplest way to reconstitute immunodeficient mice with human immune cells is through intravenous or intraperitoneal injection of HSC-CD34 cells, and immunodeficiency is a prerequisite to increase the receptivity of immunodeficient/transgenic mice for human cells or tissues engraftment and repopulation [189]. The extent of immunodeficiency influences survival and function of transplanted human cells, and thus SCID mice lacking T and B cells supported the engraftment followed by repopulation of human immune cells for several weeks [190,191]. However, due to residual innate immunity and leakiness (development of adaptive immune cells in aged mice), SCID mice did not fully support the engraftment of human cells. Reduced natural killer (NK) cells in non-obese diabetic NOD-SCID mice showed significantly improved human cell engraftment [192] due to the reduced production of IFN-γ in NOD/SCID. Profound and long-lasting impairment in adaptive and innate immunity by targeted mutation of IL-2R gamma-chain gene in NSG or BALB/c recombination activating gene (Rag)2^−/−^IL-2Rγ^−/−^ mice exhibited stable and long-term survival of transplanted human cells and tissues [193,194,195,196,197] (Figure 4).

## 7. Discussion

Different studies concluded that during the progression of IBD, T-lymphocytes exhibit a tremendous role in maintaining intestinal homeostasis and reducing tissue damage by inhibiting immune cell responsiveness with the help of Tregs [198,199]. Furthermore, defects in the number and distribution of functional Tregs and their impaired trafficking ability in the gastrointestinal tract have been examined in patients with IBD [18,104]. Recently, many studies have shown that IBD is ameliorated by restoring anti-inflammatory pathways, which mainly include increased Tregs numbers or maintaining Treg/Th17 balance by suppressing Th1/Th17 cells tipping the immune balance towards the generation of sizeable functional Tregs in IBD [21,200,201,202].

Unmet need for discovery of the novel therapeutic approaches prompt scientists to come up with alternative arrangements since many patients do not respond to treatment with approved drugs [203]. Despite the similarity between mouse and human flora, significant differences make it difficult to study all immune pathways responsible for IBD pathogenesis in one animal model. Therefore, extensive research and seamless efforts to design an ideal animal model aiming at developing a viable therapeutic strategy and selective treatment like small molecules, biologics, traditional and emerging modified therapies with minimal adverse reaction would establish the mouse-human chimeras a better platform to study autoimmune and chronic inflammatory disease. We believe humanized mouse models shall revolutionize the translational biomedical research to study inflammatory diseases.

Chemical induction, adoptive cell transfer, congenital models for IBD have been prominently utilized to study IBD pathogenesis. Further, genetically engineered murine models have been used to experimentally induce colitis to understand the mechanisms underlying intestinal inflammation and preclinical trials for developing novel therapeutic strategies [162]. Nearly 9 groups of IBD mouse models are developed based on gene targeting strategy which includes conventional, cell-specific, inducible, conventional, cell-specific transgenic, dominant-negative, mutagen induced, knock-in and innate [162]. Dysregulated innate and adaptive immune responses drive about 40 different immune-specific KO mice that spontaneously induce intestinal inflammation. The defective mucosal barriers, deregulated necroptosis/apoptosis, antibacterial peptide depletion and endoplasmic reticulum stress induce colitis spontaneously in 18 intestinal epithelial cell-specific KO mice. Although 74 types of genetically engineered mouse strains [163] and 160 IBD susceptible genes in humans indicating complex mechanism of IBD and diverse disease conditions, IBD remains elusive and needs to be studied. Out of 140 susceptible genes responsible for CD and UC, only a few had been identified as NOD2 CD-specific genes [163]. NOD2-KO and Atg16L1 mutant mice comprise of IL-10RA or IL-10RB mutation leading to the severe onset of complicated IBD pathogenesis [204]. Goettel et al. conducted studies on human immune cells reconstituted immunodeficient mice to ameliorate colitis by the expansion of Tregs following low dose IL-2 treatment and test therapeutic molecules [180].

Relevance of efficiency of Tregs as a therapeutic regime to control IBD has been demonstrated in experimental animal models and IBD patients [14,19,22,205]. The induced inflammation and severity of colitis was subsided by the adoptive transfer of Tregs [19], and promising results were observed with the use of ovalbumin (OVA)-specific Tregs in IBD [93,155,156]. The existing therapies to address IBD are based on broad suppression of inflammation that results in variable clinical advantages and unwanted adverse effects. However, passive transfer of Trges is a potential therapeutic regime aiming at promoting immune tolerance in the developed animal models, including humanized model with reconstituted human immune system mice (HIS) that supports the expansion of Tregs. Thus, induction of Tregs generation and re-establishment of immune tolerance leading to immune homeostasis is a potential approach for the long-term treatment of IBD since this regime might minimize the deleterious side effects associated with in-use immunosuppressive approaches.

## 8. Conclusions

The vital role played in IBD is evident since functional and phenotypic defects along with the compromised numbers and functions of Tregs. These cells significantly subside the intestinal inflammation mounted by the colitis. Therefore, future research concerning IBD progression and its treatment should focus on developing new clinical approaches to increase their regulatory effects including enhancing their homing to inflammation sites, expansion, or enhancing their differentiation, stability, and tissue repair properties. Further, enhancing their survival and anti-inflammatory property to maintain immune homeostasis in the gut is an important aspect. The engineered mouse-human chimera with the repopulated human immune system might be a stepping stone towards studying IBD, its biology and pathogenesis and help developing Tregs-based newer interventional strategies for IBD (Table 2).

## Figures and Tables

**Figure 1 cells-10-01847-f001:**
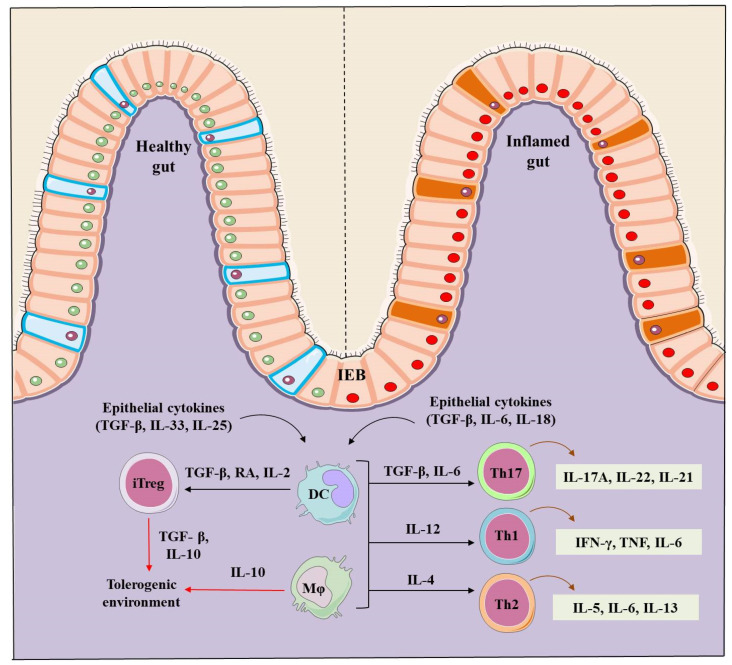
Regulation of the intestinal homeostasis in healthy and IBD inflamed gut. A healthy intestinal epithelial barrier (IEB) in presence of TGF-β, retinoic acid (RA) and IL-2 promote dendritic cells (DCs) and macrophages (mφ) to stimulate the generation of inducible Trges (iTregs). TGF-β and IL-10 are markers that contribute to the generation of iTregs, and establish and maintain the tolerogenic environment in a healthy gut. On the contrary, IBD induced inflammation induces intestinal epithelial barriers and secrete TGF-β, IL-6 and IL-8, stimulating DCs and mφ to produce the inflammatory Th-17 (IL-17A, IL-22, IL-21), Th-1 (IFN-γ, TNF-α, IL-6) and Th-2 type cells (IL-5, IL-6, IL-13) creating an inflammation focus and diseased intestine.

**Figure 2 cells-10-01847-f002:**
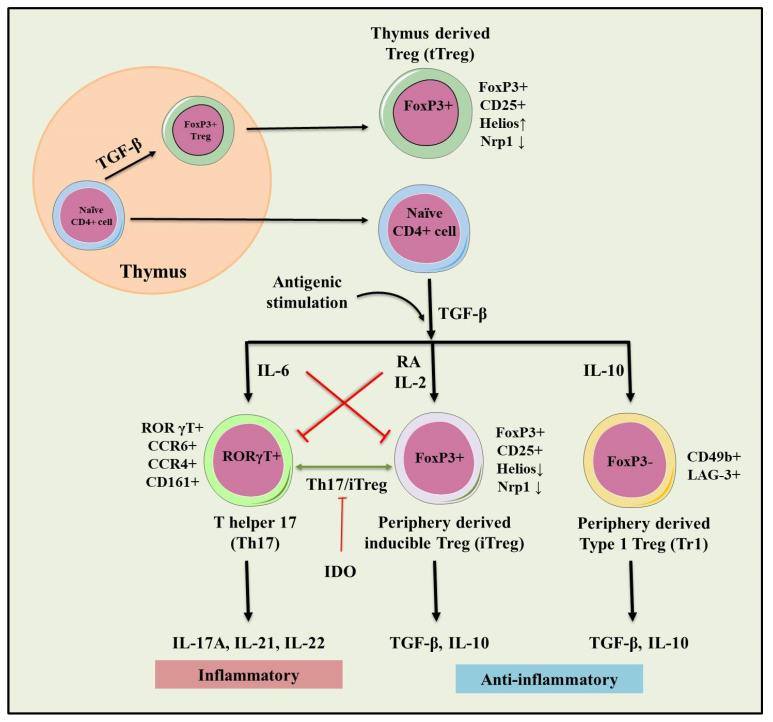
Role of natural and inducible regulatory T cells (iTregs) involved in the pathogenesis of IBD. Inflammation (IL-6) dependent interconversion of regulatory and effector T cell phenotype and role in dol 2, 3 dioxygenases (IDO) in the generation of inducible Tregs (iTregs)**.** Inflammatory (IL-17A, IL-21, IL-22) and immunosuppressive (TGF-β, IL-10) conditions following antigenic stimulation were seen during the conversion of Th17 to iTreg phenotype. This interconversion plays a crucial role in maintaining tolerance towards IBD.

**Figure 3 cells-10-01847-f003:**
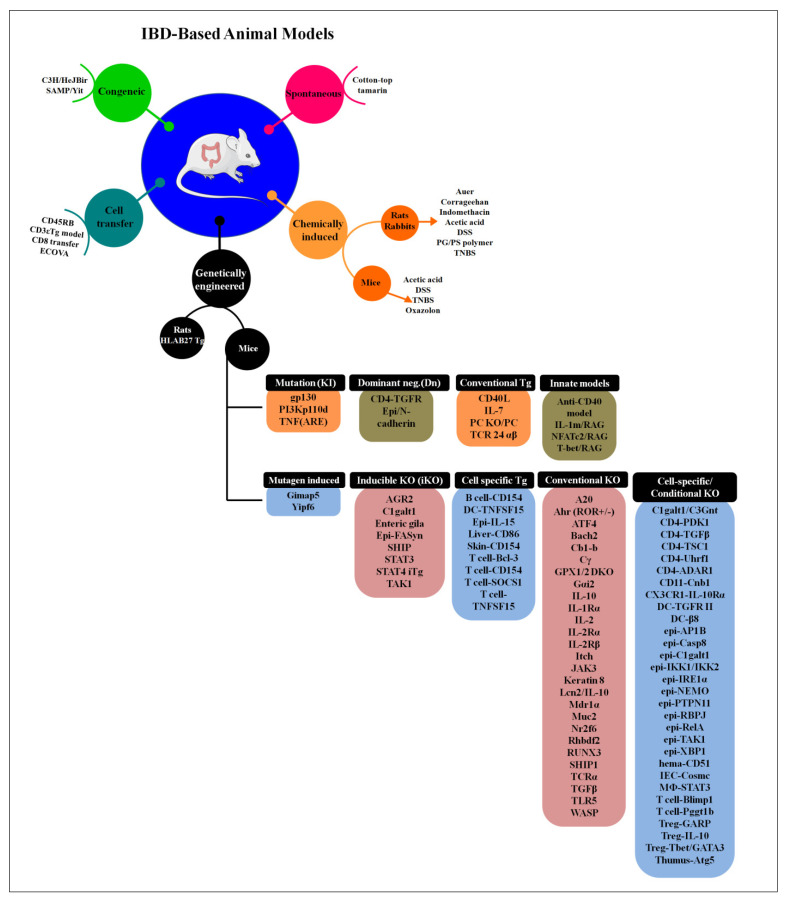
The different types of animal model used to study IBD pathogenesis. In genetically engineered type, different genes are targeting and according to it they are mainly divided into 9 different groups (the detailed information is reviewd in [162]) (adapted and modified from [157,162]).

**Figure 4 cells-10-01847-f004:**
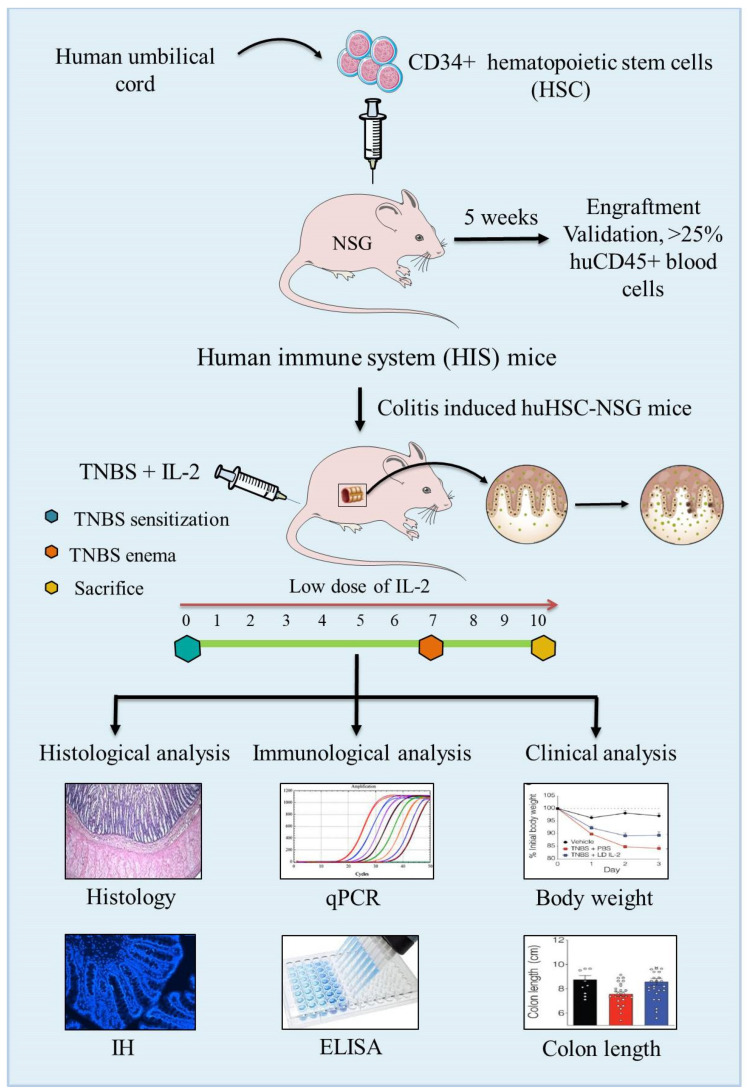
Schematic presentation of the development of the human immune system (HIS)-repopulated mice by the transplantation of CD34+ human hematopoietic stem cells (HSCs) in NOD.PrkdcscidIl2rg−/− (NSG) immunodeficient mice and the induction of experimental colitis by TNBS to assess the pathology and immunopathogenesis of colitis.

**Table 1 cells-10-01847-t001:** The different types of animal model induced through various methods are summarized with their implication and role in IBD pathogenesis ([52,162]) (the generation of different genetically engineered mice are reviewed in detailed in [162,163]).

Sr. No.	Group	Animal Model	Information
1.	Chemically Induced	Acetic acid (rat)	Reduced inflammation and myeloperoxidase activity (MPO) activity, restoration of contraction of isolated colon. Conclusion: Cyclo-oxygenase (COX) and lipo-oxygenase-mediated proinflammatory products mediated IBD pathogenesis.
		DSS	CD4+ T-cell transfer colitis model (Rag−/−) and acute DSS-induced IBD used for identifying the function of Foxp3^+^ Tregs. Conclusion: Isolated CD4+ T-cells from Foxp3^+^ Tregs-depleted mice secrete IL-13, IL-17A, and IFN-γ with severe IBD. Foxp3^+^ Tregs establish mucosal homeostasis, a therapeutic option for patients with IBD.
		TNBS	Depletes colon-specific Foxp3^+^ Tregs but with no effect on spleen, mesenteric lymph nodes, and ileum. Higher expression of Fas ligand in colitis mice; no depletion of colon-specific Tregs in DNBs-induced colitis in Fas−/− deficient mice. Conclusion: Fas/FasL pathway mediates depletion of Foxp3^+^ Tregs in the colon.
		Oxazolon	Experimental colitis induced in SJL/J mice. Th2-driven production of IL-4 and IL-5. Conclusion: Higher similarity with human UC, and Th2 response helps better understand UC.
		PG/PS polymer (Peptidoglycan polysaccharide)	Elevates plasma nitrite and nitrate levels, higher colonic mucosal permeability, and MPO activity. Conclusion: PG-PS induces chronic colitis in rats confirmed by higher NO production.
		Auer	Experimental colitis was induced by Auer with increased vascular permeability. Conclusion: Induced colitis helps better understand the injury mechanism as well as the pathogenic mechanism.
		Carrageenan (CGN)	Degrades CGN, induces ulcers in mice, rats, rabbits, and guinea-pigs. Changes spleen lymphocytes activity, suppresses immune system to cause IBD. Conclusion: CGN-based colitis follows the NF-κB signalling pathway, upregulate TNF-α and ICAM-1.
		Indomethacin	Inhibition of prostaglandin E1 and E2. Conclusion: Higher production of reactive oxygen species (ROS) and other free radicals, as well as apoptosis mediated by caspase-3, which causes IBD.
2.	Adaptive Cell Transfer	CD45RB	Naïve and memory CD4+ T-cell populations (Th1 and Th2 clone) are included in CD45RBhigh and CD45RBlow fractions. Adoptively transferred CD4+ CD45RBhigh T-cells extracted to SCID mice from wild-type mice developed colitis in 6 to 12 weeks. Conclusion: It helps unravel earlier immune–inflammation events.
		ECOVA	BALB/c and SCID mice received CD4 T-cells purified from Rag−/− mice crossed to Tg mice expressing ovalbumin (ova)-specific TCR. Conclusion: Predominant production of IL-4 in the early stage and IL-10 in the later stage in ova-specific CD4 T-cells was observed. Additionally, co-transfer of IL-10 secreting ova-specific CD4 T-cells prevented the development of colitis, and expanded ova-specific CD4 T-cells induced lymphadenopathy and caused colitis.
		CD8 Transfer	DNBS causes colitis, IFN-γ-producing cytotoxic CD8+ T-cell (Tc1) recruitment. Colitis was prevented by the antibody depletion of CD8+ and not with CD4+ T-cells. Conclusion: Relapse of colitis in normal mice with Ag-specific CD8 T-cells reveals TC1′s role in intestinal inflammation.
3.	Bacteria-Infected Model	*Citrobacter rodentium*	Model for human infectious colitis induced by *E. coli*. Transfer of CD4+ T-cells induced the secretion of IFN-γ, IL-17, IL-2 and ameliorates the activity of IL-10. Conclusion: CD4+ population generated in *C. rodentium* infection mice renders protection to the non-infected recipient via Th1-induced species-specific immune response (mainly IL-17), elevated secretion of systemic IgG, and fecal IgA.
		*Helicobactor hepaticus*	Homozygous SCID mice (with CD45RB cells) infected with *H. hepaticus* and CD4+ CD45RBhigh T-cells. Conclusion: Allows the investigation of abnormal immune response and disease.
4.	Conogenic Model	C3H/HeJBir	C3H/HeJBir (C3Bir) mice cause a missense mutation in the third exon of the Tlr4 gene, resulting in the spontaneous development of inflammation in the colon and cecum. Conclusion: Helpful in understanding the immune system and formal genetic studies of the disease.
		SAMP1/Yit	A new senescence-accelerated mouse (SAM) P1/Yit strain was established that spontaneously developed enteric inflammation under specific pathogen-free conditions. Develops CD-like ileitis, with higher levels of IFN-γ and TNF-α. Inflated levels of IL-13 and IL-5 point out the role of Th2 in causing chronic inflammation. Conclusion: Validates the role and interaction of gut microbiota in IBD pathogenesis.
		SAMP1/YitFc	Developed by mating brother–sister for over 20 generations. Conclusion: Ileitis was developed at the earlier age of 10 weeks. Useful in understanding the chronic pathological conditions of CD to help design novel therapeutic regimens.
5.	Spontaneous	Cotton-top tamarin	Small unique primate group that develops spontaneous colitis, similar to human UC. Develops secondary complications of sclerosing cholangitis, colon-based adenocarcinoma, and elevated fecal TNF-α, seen in human UC. Conclusion: Role of anti-TNF-α in human UC under investigation.

**Table 2 cells-10-01847-t002:** Developed humanized mouse models to study IBD pathogenesis.

Genetic Background of Mice	Induction of Colitis	Human Cells Transplanted	Remarks	References
NOD-SCID IL2Rγ^−/−^ (NSG)	Allergen (Birch, grass, Hazelnut)	PBMCs from allergic and non-allergic subjects	The amplified extent of colitis was seen in allergic donors isolated PBMCs engrafted micecompared to healthy donors.	[206]
NSG	Oxazolone	PBMCs from healthy, UC and AD subjects	The described model showed the potential to study the efficacy of therapeutics targeting human lymphocytes in a model closely mimicking human ulcerative colitis.	[207]
NSG	TNBS	HLA-matched human CD4+ T cells	Adoptive transfer of human CD4+ T cells in humanized animals with TNBS induced small bowel enteropathy and promoted colonic inflammation.	[208]
	HLA-matched CD34+ human HSCs from healthy and IPEX subjects		The study established the use of human HSCs to transfer disease phenotype in humanized mice to study human immune effectors and pathogenesis of IBD.	[209]
NSG	TNBS	HLA-matched human CD4+ T cells isolated from a healthy donor	The study developed an experimental humanized murine model to investigate human CD4+ T responses in vivo and identify the ITE (a non-toxic AHR agonist) as a potential therapy to achieving the immune tolerance in the intestine.	[210]
NSG	TNBS	PBMCs isolated from healthy donors	Low-dose IL-2 helped in expanding Trges for using as a therapeutic strategy against colitis in humanized mice.	[180]
NSG	PBMCs from UC donors were reconstituted in NSG mice and treated with oxelumab		NSG-UC mice treated with oxelumab significantly reduced clinical, colon and histological scores and reduced serum levels of IL-6.	[211]
NSG	TNBS	Antisense+ CD4 cells isolated from a healthy donor	Silencing the endogenous antisense long non-coding RNA restores CD39 levels with enhancing Treg-suppressive function.	[212]
NSG	TNBS	CD34+ human HSCs	Low-dose (LD) IL-2 reduced the severity of TNBS induced in HSC CD34+ reconstituted NSG mice and paved the way for developing future therapeutic strategy based on LD IL-2	[185]

Abbreviations: NOD-SCID IL2Rγ^−/−^, non-obese diabetic mice with severe combined immunodeficiency and null mutation in the interleukin 2 (IL-2) receptor gamma chain; PBMCs, peripheral blood mononuclear cells; UC, ulcerative colitis, AD, atopic dermatitis; TNBS, tri-nitro-benzene-sulfonic acid; HSCs, hematopoietic stem cells; IPEX, immuno-dysregulation polyendocrinopathy enteropathy X-linked syndrome; ITE,2-(1′H-indole-3′-carbonyl) thiazole-4-carboxylic acid methyl ester; AHR, aryl hydrocarbon receptor.

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
