# Peer review of "Translating Treg Therapy for Inflammatory Bowel Disease in Humanized Mice"

_cells, 2021, doi:10.3390/cells10081847_

Round 1
Reviewer 1 Report
The manuscript is interesting and novel. It brings a significant update on the subject matter. Following need to be addressed before acceptance. It requires revision.
The comments are following:
1. Can the authors describe which model is best for the study of Treg hypothesis. Its better if the authors can summarize the merits/applicability of numerous models in respect to testing of Treg hypothesis. In this regard Table 1 can be modified to summarise better the chemical and genetic models.
2. A summary table of available agents of natural or synthetic origin need to be included, whether seprately or together depends on the number of agents.
3. Impact on disease activity index should be highlighted and included in the scheme.
4. What is the state of agents evaluated till date in the models.
5. A scheme on proposed mechanism of the agents targeting Treg can be included.
6. In discussion or elsewhere the benefits of Treg over existing agents can be summarized.
Author Response
Reviewer 1
The manuscript is interesting and novel. It brings a significant update on the subject matter. Following need to be addressed before acceptance. It requires revision.
Comments 1: Can the authors describe which model is best for the study of Treg hypothesis. It’s better if the authors can summarize the merits/applicability of numerous models in respect to testing of Treg hypothesis. In this regard Table 1 can be modified to summarize better the chemical and genetic models.
Response: We are thankful to the reviewer for their valuable time and critical comments. In the revised manuscript we have discussed about the importance of humanized model and expansion of Treg cells (Section 7, Discussion). We have also added the detailed information about various animal-based model used for IBD and their role in table 1.
Comments 2: A summary table of available agents of natural or synthetic origin need to be included, whether separately or together depends on the number of agents.
Response: We are grateful the reviewer for helping us making our manuscript more approachable to the readers. In the revised manuscript we have included various agents used for developing IBD experimental model (table 1).
Comments 3: Impact on disease activity index should be highlighted and included in the scheme.
Response: We are grateful to the reviewer for providing us an opportunity to revise the manuscript. We have modified the schematic presentation for better understanding. We hope this meet the standards.
Comments4: What is the state of agents evaluated till date in the models.
Response:While we appreciate comments of our fellow reviewer, we want to inform s/he that this manuscript focuses on the Treg as therapeutic regime in HSC CD34 reconstituted NSG mice. Still respecting our fellow learned reviewer’s comments, we have enlisted the list of chemicals used to induce experimental colitis in animals. Please see the revised manuscript with changes marked/highlighted.
Comments 5: A scheme on proposed mechanism of the agents targeting Treg can be included.
Response: I think our fellow reviewer wants to say that mechanism wherein Tregs can be used to treat IBD/colitis. We are not targeting Tregs, instead using this cell type as a therapeutic regime to treat and alleviate colitis. Pls see the modified figure 3.
Comments6: In discussion or elsewhere the benefits of Treg over existing agents can be summarized.
Response: We acknowledge the reviewer for the constructive comments. In the modified manuscript we have discussed about the benefits of Treg over existing therapies (Section 7, Discussion).
Reviewer 2 Report
Review:
This is an interesting review about the “*Translating Treg therapy for
inflammatory bowels disease in* *humanized mice.”*
The manuscript is well written, and I would like to highlight the
usefulness of the Figures.
They are some comments to make on this publication, which would suit
perfectly to the readership of the Cells.
1. There are a few typo, please revise and correct them, start with the
title (bowel) (for example line 40, 90, 160, 197…373…)
There are some hardly understandable sentences, please revise
2. The therapeutic role in IBD section suggested to revise by an
IBDologist! Anti-TNF agents are not associated with higher cancer rate in
general.
European Evidence-based Consensus: Inflammatory Bowel Disease and
Malignancies
Vito Annese, et al. *Journal of Crohn's and Colitis*, Volume 9, Issue 11,
November 2015, Pages 945–965, https://doi.org/10.1093/ecco-jcc/jjv141
*ECCO Statement 5A:* There is currently no evidence that the overall risk
of cancer is increased in patients being treated with anti-TNF agents alone
(NMSC risk is elevated)
3. 92 reference is irrelevant there.
4. 93 reference need to be updated.
5. 94-97 references need to be updated, we have new data on efficacy and
safety of these treatments.
6. Line 259 needs to be edited
7. Please use abbreviations only after the first use of the whole
expression (in some cases the explanations is later – it’s understandable,
but not official)
8. Section 9 should be shorted
9. 458 line irrelevant references
10. Please use the journal reference format
Journal Articles:
1. Author 1, A.B.; Author 2, C.D. Title of the article. *Abbreviated
Journal Name* *Year*, *Volume*, page range.
11. Please use the official titles such as Discussion and Conclusions
Author Response
Reviewer 2
This is an interesting review about the “*Translating Treg therapy for inflammatory bowels disease in* *humanized mice.”*The manuscript is well written, and I would like to highlight the usefulness of the Figures.
Comments 1: There are a few typo, please revise and correct them, start with the
title (bowel) (for example line 40, 90, 160, 197…373…). There are some hardly understandable sentences, please revise.
Response:Authors appreciate the kind words scribbled by the reviewer on our work and we are thankful to thereviewers for theappreciation.The manuscript has been thoroughly read and corrected following reviewersrecommendations. We believe the revised manuscript meets the journal’s standard and is in good shape for publication in Cells.
Comments 2: The therapeutic role in IBD section suggested torevise by an
IBDologist! Anti-TNF agents are not associated with higher cancer rate in
general.European Evidence-based Consensus: Inflammatory Bowel Disease and
Malignancies. Vito Annese, et al. *Journal of Crohn's and Colitis*, Volume 9, Issue 11,
November 2015, Pages 945–965, https://doi.org/10.1093/ecco-jcc/jjv141. *ECCO Statement 5A:* There is currently no evidence that the overall risk
of cancer is increased in patients being treated with anti-TNF agents alone
(NMSC risk is elevated)
Response: We appreciate the critical evaluation by the reviewer. We have gone through the reference suggested by the reviewer and we agree with the point that cancer risk is not elevated with anti-TNF agents, however; anti-TNF therapy for RA is associated with an increased risk of NMSC.
As per the suggestion we have modified the statement. We regret the confusion caused.
Comment 3: 92 reference is irrelevant there.
Response:The necessary correction was done and appropriate reference has been cited.
Comment 4: 93 reference need to be updated.
Response: The respective reference was update with other information and latest references.
Comment 5: 94-97 references need to be updated, we have new data on efficacy and
safety of these treatments.
Response: The necessary correction was done with the latest information by citing new references.
Comment 6: Line 259 needs to be edited.
Response:We have modified the sentence for better understanding.
Comment 7: Please use abbreviations only after the first use of the whole
expression (in some cases the explanations is later – it’s understandable,
but not official).
Response:We have modified the manuscript for the same.
Comment 8: Section 9 should be shorted
Response:As per the suggestion we have modified the segment.
Comment 9: 458 line irrelevant references
Response:The necessary correction was done and cited the appropriate reference.
Comment 10: Please use the journal reference format
Journal Articles:
1. Author 1, A.B.; Author 2, C.D. Title of the article. *AbbreviatedJournal Name* *Year*, *Volume*, page range.
Response:We have tried our best to resolve the issue by following the MDPI specific guidelines.
Comment 11: Please use the official titles such as Discussion and Conclusions
Response:In the modified manuscript we have used official titles. We thank the reviewers for shaping our manuscript in a better way.
Round 2
Reviewer 1 Report
There is too much information in table 1. Please summarize it in bullet points.
Author Response
Comment: There is too much information in table 1. Please summarize it in bullet points. Response: Although Table 1 is inclusive and conveys a whole lot of information, respecting our fellow reviewer's views we have shortened Table 1 and is quite good for citing in our manuscript.